# Automatic Seamline Determination for Urban Image Mosaicking Based on Road Probability Map from the D-LinkNet Neural Network

**DOI:** 10.3390/s20071832

**Published:** 2020-03-26

**Authors:** Shenggu Yuan, Ke Yang, Xin Li, Hongyue Cai

**Affiliations:** 1China Transport Telecommunications and Information Center, Beijing 100011, China; shengguyuan@whu.edu.cn; 2Department of Systems Design Engineering, University of Waterloo, Waterloo, ON N2L 3G1, Canada; 3Guojiao Spatial Information Technology (Beijing) Co., Ltd., Beijing 100011, China; lixin_kj@cttic.cn (X.L.); caihongyue@cttic.cn (H.C.)

**Keywords:** mosaicking, urban image, seamline determination, deep learning, D-LinkNet

## Abstract

Image mosaicking which is a process of constructing multiple orthoimages into a single seamless composite orthoimage, is one of the key steps for the production of large-scale digital orthophoto maps (DOM). Seamline determination is one of the most difficult technologies in the automatic mosaicking of orthoimages. The seamlines that follow the centerlines of roads where no significant differences exist are beneficial to improve the quality of image mosaicking. Based on this idea, this paper proposes a novel method of seamline determination based on road probability map from the D-LinkNet neural network for urban image mosaicking. This method optimizes the seamlines at both the semantic and pixel level as follows. First, the road probability map is obtained with the D-LinkNet neural network and related post processing. Second, the preferred road areas (*PRAs*) are determined by binarizing the road probability map of the overlapping area in the left and right image. The PRAs are the priority areas in which the seamlines cross. Finally, the final seamlines are determined by Dijkstra’s shortest path algorithm implemented with binary min-heap at the pixel level. The experimental results of three group data sets show the advantages of the proposed method. Compared with two previous methods, the seamlines obtained by the proposed method pass through the less obvious objects and mainly follow the roads. In terms of the computational efficiency, the proposed method also has a high efficiency.

## 1. Introduction

Orthoimages have increasingly become a popular visualization product and planning instrument for integrating the rich information content of images with the geometric properties of maps (ground projection) and can be easily combined with additional information from geographic information systems (GIS) to create an orthoimage map [1]. However, with the development of technology, the orthoimage spatial resolution becomes higher, and the coverage area of an individual orthoimage is typically very small Thus, image mosaicking is a necessary process of constructing multiple images into a large-scale and single seamless composite image. This process has been applied in a wide variety of applications such as environmental monitoring, agricultural monitoring, and disaster management [2,3]. Orthoimages are typically orthorectified by the Digital Terrain Model (DTM) of the same geographical area. The quality of DTM directly affects the accuracy of orthorectification. Objects not contained in the DTM cannot be orthorectified correctly. Those objects would appear at different locations in the overlapping area and cause visual discontinuities in image mosaicking. An ideal seamline should avoid such objects [1,4,5,6,7,8]. A seamline is the line along which overlapping areas will be mosaicked. Each pixel in the final mosaicking result is represented entirely by only one orthoimage based on which side of the seamline it lies on. Seamline is also helpful when overlapping areas have significant differences in features. When mosaicking orthoimages, seamline determination is one of the most difficult technologies for compositing a single seamless orthoimages. The purpose of seamline determination is to find the seamlines with the minimal intensity and gradient differences in the overlapping area. In this paper, our work focuses on automatic seamline determination for urban image mosaicking.

In order to minimize the transition of the final mosaic image, the ideal seamline should avoid crossing obvious objects as much as possible and go along the objects which have small relief displacement. Differential expression is essential for seamline determination, which is a measure of the difference between left and right image overlap areas [1,7]. The method based on differential expression is the basic method of seamline optimization. The method first measures the difference between the overlap region images to form a difference matrix, and then uses the path search algorithm to obtain the final optimized seamline. According to the differential expression algorithm and the path search algorithm, the recently proposed seamline determination algorithms are as follows.

Milgram [9] defined the “best” seamline point for each line of the overlapping area that minimizes the sum of the gray differences between the left and right images. Afek and Brand [10] integrated global feature matching and local transformation into seamline determination. Soille [11] used the mathematical morphology and marker-controlled segmentation paradigm to determinate the seamlines. The difference (geometric and radiometric discontinuities) can be minimized if the seamlines go along salient image structures.

Kerschner [1] proposed a “two snake” method for seamline determination. The main idea is to design a double snake model, through mutual attraction of the two snake lines, and finally form a snake line to obtain an optimized seamline. The energy of the double snakes is defined based on similarity. The seamlines go along the region of maximum similarity. The criteria for regions of similarity are color similarity (hue and intensity) and texture similarity (orientation and magnitude of gradients). Wang et al. [12] proposed a seamline determination in image mosaicking using improved snakes. The integrated snake model and Bresenham algorithm was presented, which the Bresenham algorithm was used to calculation the photometric. This solves the local optimum problem that exists in the snake model to some extent, but not completely.

Ma and Sun [13] proposed a seamlines optimization for image mosaicking with airborne light detection and ranging (LiDAR). According to the raw laser scanning dataset, the high ground objects of the overlap area were identified as the obstacle. Then, the A* algorithm was used to determine the final seamlines, and the seamlines were kept away from these obstacles in the registered images. Wang and Wan [14,15,16] presented a seamline determination with the aid of vector roads for the first time. In this approach, firstly, with the help of the vector roads and their widths, the seamlines will go along the centerlines of roads with a large width as much as possible and avoid crossing the obvious objects. Finally, the shortest path algorithm is applied to determine the final seamline. Chen et al. [17] first used the Digital Surface Model (DSM) and Digital Terrain Model (DTM) to derive an Orthoimage Elevation Synchronous Model (OESM) that accurately reflected the pixel of each digital orthophoto image, and then obtained the final optimized seamline using the Dijkstra algorithm. Wang et al. [2] used vector building maps to determine the seamlines, which guaranteed the seamlines avoiding the crossing of buildings as much as possible. Different from the method of tracking vector roads, the seamlines determined by this method went along the middle line between buildings in order to avoid crossing the obvious objects, especially for high-rise buildings.

Using the normalized difference vegetation index (NDVI) and morphological building index (MBI), Pan et al. [18] introduced ground object classification into the seamline optimization method. In this approach, based on ground object classes, three types of areas, that is, obstacle areas, preferred areas, and general areas, are further formed. Then, each type of region is assigned a different weight to optimize obtained pixel-size costs. Finally, Dijkstra’s algorithm is carried out to search the shortest path as the final seamlines based on previously determined pixel-size costs in overlapping areas.

Chon et al. [19] first sought the maximum difference by minimizing the maximum, and then used Dijkstra’s algorithm to determine the final seamlines. The method, which is based on minimizing the maximum difference, measures the difference by a local region. It first calculates the Normalized Cross Correlation (*NCC*) of the central pixel in this window, and makes the contrast between the normalized correlation coefficients by exponential stretching of further expansion. Then, using the strategy of minimizing the maximum difference on the basis of the difference matrix, the seamline is prohibited from crossing the region, and finally the final optimized seamline is obtained by the Dijkstra algorithm.

Pan et al. [6] used image segmentation to determine seamlines for orthoimage mosaicking in an urban area for the first time. This method uses segmentation to improve seamline determination. Firstly, the preferred regions were selected according to the spans of objects segmented by the mean shift algorithm. Then, Dijkstra’s shortest path algorithm was adopted to determinate the final seamline. Since this method was proposed, several object-based methods have been used to optimize seamlines. After that, Pan et al. [20] introduced the region change rate (RCR) into seamline optimization for orthoimage mosaicking. The change rate of the regions acquired by the mean shift segmentation algorithm were defined by the percentage of the changed pixels. This method determined the seamlines at object-level and pixel-level. Wang et al. [7] adopted watershed segmentation for seamline optimization at both the object and pixel level. Using normalized cross correlation, the obvious objects, such as buildings, were excluded from the preferred objects areas at the object level. Dijkstra’s algorithm found the final seamlines in the preferred objects areas at the pixel level.

Li et al. [21,22] first adopted the graph cuts energy minimization framework to find the optimized seamlines. The image color, gradient magnitude and texture were combined in the smooth energy functions in the graph cuts energy minimization framework. The determined seamlines passed through the areas of a smooth texture, such as roads, woodlands, and green lands. Li et al. [23] proposed an automatic seamline optimization based on graph cuts in UVA image mosaicking. 

Yuan et al. [24] proposed a seamline optimization based on the disparity image by the semi-global matching (SGM) algorithm. After obtaining the disparity image, the mathematical morphology method was employed to deal with the noises and small holes of the disparity image in order to determine the non-ground area. Finally, an improved greedy snake algorithm was adopted for the final seamlines. Similar to that algorithm, Pang et al. [25] introduced dense matching into seamline determination. Firstly, the SGM was used to estimate the disparity of each pixel. Next, the obstacle and non-obstacle areas were determined by a predefined threshold. Finally, Dijkstra’s algorithm was adopted to optimize the final seamlines in avoiding crossing the obstacle area as much as possible.

Based on the integrated deep convolutional neural network (CNN) and graph cuts energy minimization framework, Li [26] proposed a novel algorithm to optimize seamlines for image mosaicking. Different from the previous method [22], this method defined similarity energy terms of the graph cut using the semantic classification classified by the CNN instead of using the color, gradient, or texture.

In our paper, we propose a novel method of seamline determination based on a road probability map which is extracted by the D-LinkNet neural network for urban image mosaicking. This method optimizes the seamlines at both the semantic and pixel level. Firstly, the D-LinkNet neural network is adopted to obtain the road probability map of the overlapping area in the left and right image respectively. Secondly, the preferred road areas (*PRAs*) are determined by binarizing the road probability map of the overlapping area both in the left and right image. The *PRAs* are priority areas which the seamlines cross. Finally, the final seamlines are determined by Dijkstra’s shortest path algorithm implemented with binary min-heap at pixel level.

The remainder of this paper is organized as follows: Section 2 describes the proposed seamline determination for urban image mosaicking based on a road probability map from the D-LinkNet neural network, where Section 2.1 introduces road probability map generation by D-LinkNet. Section 2.2 presents the determination of *PRAs,* and Section 2.3 introduces pixel-level seamline determination; Section 3 describes the experimental results and analysis, where Section 3.1 presents the experimental data and platform and Section 3.2 presents the seamline determination results and analysis. Section 4 draws the conclusions.

## 2. Materials and Methods 

The major difficulty issue of seamline determination is to define the differential expression of the overlapping area more accurately. A cost image is generally adopted to express the difference of the overlap image. Most seamline determination methods use the pixel-by-pixel or local regular subimages to define the differential expression, and it is difficult to measure the difference accurately [6,7]. Object recognition is considered to be helpful for differential expression. If object recognition has been solved perfectly, we can set the areas with the high differential expression highest cost, such as buildings. Then, the seamlines can be guaranteed not to go across stand-alone objects such as buildings. However, object recognition is a complicated problem [6]. With the help of a vector roads network, some approaches have used vector roads to optimize the seamlines, in which the seamline follows the centerlines of roads with a large width as much as possible and avoids crossing the obvious objects. Such seamlines are benefited to maintain the integrity of objects and improve the quality of image mosaics [14,15,16]. Based on this idea, we utilized a road probability map from the D-LinkNet neural network to optimize the seamline determination.

The process flow for the proposed method is shown in Figure 1. The proposed algorithm optimizes the seamlines at the semantic and pixel level. The D-LinkNet neural network is adopted to achieve the road probability map of the overlapping area both in the left and right image. At the semantic level, the *PRAs* are determined by binarizing the road probability map. In this step, most of the roads will be included in the *PRAs*. The *PRAs* are the priority areas which the seamlines cross. At the pixel level, the Dijkstra’s shortest path searching algorithm is adopted to find the final seamlines.

### 2.1. Road Probability Map Generation by D-LinkNet

#### 2.1.1. Block Road Probability Map Generation by D-Linknet

Road extraction from high-resolution images is a basic application of remote sensing, which has attracted the attention of both academics and industry for a long time [27,28]. With the development of deep learning and contribution of specialized datasets from the remote sensing community, convolutional neural networks (CNNs) have been broadly used as alternatives to traditional methods for visual recognition tasks in remote sensing, including building detection [29], road segmentation [30,31,32], and topological map generation [33]. In our research, we focus on the main part of road extraction—road probability map generation—and model it with a convolutional neural network.

D-LinkNet [34] is adopted as a basic CNN model of the proposed method due to its excellent performance in 2018’s DeepGlobe road extraction challenge and broad use as the baseline in road segmentation tasks [35]. It uses the typical encoder-decoder architecture inherited from LinkNet and adds the delated convolution part to acquire and ensemble multi-scale features to enlarge the receptive field, which is able to handle a road’s properties, such as connectivity, complexity, and long span, to some extent [36]. Specifically, ResNet34 [37] is pretrained on ImageNet [38] and used as the encoder part of D-LinkNet, and several dilated convolution layers with skip connections are placed in the center part to enhance the reception ability. The decoder part uses transposed convolution [39] layers to conduct upsampling, restoring the resolution of the feature map from one that is downsampled to the original one. The architecture of the D-LinkNet is presented in Figure 2.

In practice, considering the large size of satellite and aerial images, clipping is necessary to generate image patches with a proper and fixed size (e.g., 1024 × 1024), to make sure that D-LinkNet implementation works under a constrained computation ability and the output of separated patches has been integrated into the final result. In addition, a small size of overlap between neighboring patches should be considered and part of the redundant output within overlap areas should be discarded, as misclassification often happens at the border pixels of a patch given that the receptive field is constrained by the edge. As for the training step of deep learning, we utilized transfer learning to accelerate the whole train process, and data augmentation of the DeepGlobe road dataset [35] to promote the network’s learning ability.

Some road probability map generation by D-LinkNet is shown in Figure 3. Each size of the sample image is 1024 × 1024 pixels. The lighter the gray value of the pixel value in the image, the higher the probability that it will be recognized as a road.

#### 2.1.2. The Post Processing of Road Probability Map Generation

When processing high-resolution remote sensing images, due to the limitations of memory and other factors, block processing is required to extract the roads using D-Link. If the block extraction results are directly stitched together, this may lead to obvious visible transitions. The stitched result is shown in Figure 4a. There are obvious visible transitions in the stitched result. Figure 4a is the result of directly stitching the image according to the size of 1024 × 1024 pixels. In order to eliminate obvious visible transitions, this paper ensures a certain overlap between adjacent blocks during blocking. The overlap provides a foundation for the subsequent elimination of obvious visible transitions. When blocking, the width of the overlapping area is 511 pixels.

After processing according to the blocking principle to obtain the road extraction result, the post processing can be performed to eliminate obvious visible transitions on both sides of the seamline. In this paper, there are only vertical and horizontal seamlines. This kind of processing is performed on the two sides of the stitching point line by line (or row by row) within the artificially specified width range (the width of the range must be smaller than the width of overlap region). The method used is as follows:(1)$$\begin{array}{l} OI_{L}^{i}=O_{L}^{i}+(O_{L}^{i}-O_{R}^{i})K\\ OI_{R}^{i}=O_{R}^{i}+(O_{R}^{i}-O_{L}^{i})K\\ K=i/W\;\;\;0\leq i\leq W-1 \end{array}$$
where $O_{L}^{i}$ is the gray value of the pixel in the left (top) road probability map, $O_{R}^{i}$ is the gray value of the pixel in the right(bottom) road probability map, $OI_{L}^{i}$ is the processed gray value of the pixel in the left(top) road probability map, and $OI_{R}^{i}$ is the processed gray value of the pixel in the right(bottom) road probability map. *W* is the width of the smooth area, and *K* is the weight [40]. 

In this process, the gray values of the two images to be stitched are weight averaged pixel by pixel to be used as the gray values after stitching. The weights used vary linearly and inversely within the calculation range. This process can basically eliminate the obvious difference near the stitched line, which is shown in Figure 4b. Compared with Figure 4a, there are no obvious visible transitions in Figure 4b.

### 2.2. Preferred Road Areas Determination

After the generation of the road probability map in the overlapping area in both the left and right image, the accuracy of detecting preferred road areas is determined by the road probability threshold for binarization. In order to estimate the road probability threshold adaptively, we used the Otsu’s method [41,42] to estimate the value of the road probability threshold. Otsu’s method is used to perform automatic image thresholding. The algorithm returns a single intensity threshold that separates pixels into two classes: foreground and background. This threshold is determined by minimizing the intra-class intensity variance, or equivalently, by maximizing the inter-class variance. We estimated the road probability threshold in the overlapping in the left and right image respectively. The method used is as follows Equations (2)–(4):(2)$$I_{L}(x,y)\in PRALs\;\{\begin{array}{cc} true & P_{L}(x,y)>=T1\\ false & otherwise \end{array}$$
(3)$$I_{R}(x,y)\in PRARs\;\{\begin{array}{cc} true & P_{R}(x,y)>=T2\\ false & otherwise \end{array}$$
(4)$$I(x,y)\in PRAs\;\{\begin{array}{cc} true & I_{L}(x,y)\in PRALs\;and\;I_{R}(x,y)\in PRARs\\ false & otherwise \end{array}$$
where $I_{L}(x,y)$ is the pixel in the road probability map of the overlapping area in the left image. $I_{R}(x,y)$ is the pixel in the road probability map of the overlapping area in the right image; $P_{L}(x,y)$ is the value of $I_{L}(x,y)$; $I_{R}(x,y)$ is the value of $I_{R}(x,y)$; I(x,y) is the pixel of the overlapping area; PRALs represents the preferred road areas of the overlapping area in the left image; and PRARs represents the preferred road areas of the overlapping area in the right image. PRAs represents the preferred road areas. T1 and T2, which are estimated by Otsu’s method, are the road probability threshold of the overlapping area in the left image and right image, adaptively.

### 2.3. Pixel-Level Seamline Determination

#### 2.3.1. Pixel-Level Cost Determination

After the determination of the *PRAs,* pixel-level seamline determination is used to determine the final seamlines. The two intersecting pixels of the image borders are confirmed as the start and end points [19]. 

Similar to Chon et al. [19], the Normalized Cross Correlation (*NCC*) is adopted to quantify the difference between the overlapping area of two images at the pixel level. Considering the efficiency, the quick *NCC* is calculated with Equation (5) using the 5×5 windows. i and j are coordinates in the image coordinate system. $I_{L}(i,j)$ and $I_{R}(i,j)$ are the gray values of the overlapping area in the left and right image at (i,j), respectively. The cost value is computed in Equation (6), which has a range of 0–1.0.
(5)$$QNCC(x,y)=\frac{\sum_{i=x-2}^{x+2}\sum_{j=y-2}^{y+2}I_{L}(i,j)I_{R}(i,j)-\frac{1}{25}\sum_{i=x-2}^{x+2}\sum_{j=y-2}^{y+2}I_{L}(i,j)\sum_{i=x-2}^{x+2}\sum_{j=y-2}^{y+2}I_{R}(i,j)}{\sqrt{[\sum_{i=x-2}^{x+2}\sum_{j=y-2}^{y+2}I_{L}{\left(i,j\right)}^{2}-\frac{1}{25}{\left(\sum_{i=x-2}^{x+2}\sum_{j=y-2}^{y+2}I_{L}(i,j)\right)}^{2}][\sum_{i=x-2}^{x+2}\sum_{j=y-2}^{y+2}I_{R}{\left(i,j\right)}^{2}-\frac{1}{25}{\left(\sum_{i=x-2}^{x+2}\sum_{j=y-2}^{y+2}I_{R}(i,j)\right)}^{2}]}}$$
(6)cost(x,y)=0.5−0.5×QNCC(x,y)

The final pixel-level cost of pixel (x,y) in the overlapping area between images is defined as:(7)$$DE(x,y)=\{\begin{array}{cc} w\times cost(x,y) & I(x,y)\in PRAs\\ cost(x,y) & otherwise \end{array}$$
where I(x,y) is the pixel of the overlapping area in the left and right image. If I(x,y) belongs to *PRAs*, the cost value should be multiplied by *w*. *w* is the weight for pixels in *PRAs*, which is assigned a value much lower than 1.0. With such weight processing, this makes sure that the difference in the road area can be relatively small, so the seamlines will pass through roads as much as possible. 

#### 2.3.2. Shortest-Path Searching

After the final pixel-level cost determination, similar to Pan et al. [6], in order to minimize the difference of the seamlines, the proposed method uses the differential cost to calculate the local cost between neighboring pixels when applying Dijkstra’s algorithm to search for the shortest path. The differential cost is defined in Equation (8).
(8)$$de_{mn,pq}=\left|DE(m,n)-DE(p,q)\right|$$
where (m,n) and (p,q) are adjacent pixels; DE(m,n) and DE(p,q) are the pixel-level costs of pixels (m,n) and (p,q), respectively, which are calculated in Equation (7). Let near(m,n) be the eight neighboring nodes of (m,n), DCost(m,n), and DCost(p,q) be the global minimum costs from the start pixel to (m,n) and (p,q), respectively. Then:(9)$$DCost(m,n)=\min\{de_{mn,pq}+DCost(p,q);(p,q)\in near(m,n)\}$$

Dijkstra’s algorithm is a classic global optimization method which solves the single-source shortest-path problem for arbitrary directed graphs *G = (V, E)* with unbounded non-negative weights [43,44]. Given a source vertex s in a weighted directed graph *G = (V, E)* where all edges are nonnegative, the pseudo-code for Dijkstra’s algorithm is presented in Algorithm 1. Dijkstra’s algorithm uses a data structure for storing and querying partial solutions sorted by distance from the start. The computational complexity of the original Dijkstra’s algorithm is Θ(|V|⋅|V|+|E|), if the min-priority queue is implemented by an ordinary linked list. |V| is the number of nodes in the graph and |E| is the number of edges in the graph. The computational complexity depends on how to the min-priority queue is implemented. In order to improve the efficiency of Dijkstra’s algorithm, similar to Wang et al. [7], the proposed method implements the min-priority queue with a binary min-heap. The pseudo-code for Dijkstra’s algorithm with a binary min-heap is presented in Algorithm 2. The computational complexity of the improved Dijkstra’s algorithm is Θ((|V|+|E|)⋅lg|V|) [7].

**Algorithm 1** Dijkstra’s algorithm**1****Dijkstra**(*G*, *s*)**2**dist[*s*] = 0**3****for each** vertex *v*∈*V***4**  **if**
*v* ≠ *s***5**    dist[*v*] = ∞**6**    pre [*v*] = undefined**7***S* = *Ø***8***Q* = *V***9****while***Q* ≠ *Ø* do**10**  u = extract_min(*Q*)**11**  *S* = *S*∪{*u*}**12**  **for each** vertex *v*∈Adj(*u*) do**13**    dist[*v*] = min(dist[*v*], dist[*u*]+*w*(*u*, *v*))**14**    pre[*v*] = *u*

**Algorithm 2** Dijkstra’s algorithm with Binary Min-heap**1****Dijkstra_Binary_Min-heap**(*G*, *s*)**2**dist[*s*] = 0**3****for each** vertex *v*∈*V***4**  **if**
*v* ≠ *s***5**    dist[*v*] = ∞**6**    pre [*v*] = undefined**7**  *Q*.add_with_min-priority(*v*, dist[*v*])**8***S* = *Ø***9***Q* = *V***10****while***Q* ≠ *Ø* do**11**  u = extract_min_with_min- priority(*Q*)**12**  *S* = *S*∪{*u*}**13**  **for each** vertex *v*∈Adj(*u*) do**14**    dist[*v*] = min(dist[*v*], dist[*u*]+*w*(*u*, *v*))**15**    pre[*v*] = *u***16**    *Q*.decrease_min-priority(*v*, dist[*v*])

## 3. Experimental Results and Analysis

### 3.1. Experimental Data and Platform

The experiment consists of two parts. The first part uses the D-LinkNet neural network to generate the road probability map. The second part determines the seamline based on this road probability map.

The D-LinkNet neural network was trained and tested with a single NVIDIA GeForce GTX 1080Ti using the TensorFlow library in python in Linux. The training set was composed of 6226 images from the DeepGlobe Road Extraction dataset [35] and 1000 image sets of data for manually marking roads with 0.5-m resolution remote sensing images. The image size was 1024 × 1024. Details for training a D-LinkNet-34 network are as follows: *Batchsize* = 1, *epoch* = 200, *train_best_loss* = 50, *learning_rate* = 2 × 10^−4^, and the loss function defined as a mixture of binary cross entropy loss representing the error between pixels with a dice coefficient loss suitable for the error between batches based on the IOU(intersection over union) as an evaluation index. The output of the neural network is the probability of judging whether the pixel belongs to a road. It took almost 50 h to complete the network training. Once the network training is completed, the network can be applied to other datasets of different areas. After training, the error train loss is 0.18796 and the accuracy rate is 0.81204.

The proposed determination of the seamline method was implemented by C++ programming on a portable computer with four Intel(R) Core(TM) i7-6700HQ CPU at 2.60 GHz, 16.0 GB of internal memory, and a mechanical hard drive with a 1 TB capacity, a 32 MB cache, and a 7200 r/min speed for data processing. The Geospatial Data Abstraction Library (GDAL), which is a widely used open source library, was adopted to read and write a large variety of raster spatial data formats. *w* was the weight for pixels in *PRAs* which is shown in Equation (7). The proposed method was performed with *w* = 0.001. In our practice, the value of w was determined by our experience.

In order to verify the algorithm proposed in this paper, three sets of aerial images were selected for experiments. An overview of the three sets of data is shown in Table 1. Among them, the coverage of Dataset A is the central urban area of a large city; the coverage of Dataset B is the suburb of a big city, and the coverage of Dataset C is the center of a medium-sized city.

### 3.2. Seamline Determination Results and Analysis

In order to compare the effect and efficiency of the algorithm of our proposed approach, Dijkstra’s algorithm [43] and the OrthoVista method (INPHO, 2005) were selected to compare with the proposed algorithm. Dijkstra’s algorithm is one of the earliest and simplest algorithms. OrthoVista is one of the most widely used professional mosaicking products in the world, which is a desktop software of INPHO’s digital photogrammetric system [45].

Figure 5 illustrates the experiments of *PRAs* determination and the intermediate results for Dataset A. Figure 6 illustrates the experiments of *PRAs* determination and the intermediate results for Dataset B. Figure 7 illustrates the experiments of *PRAs* determination and the intermediate results for Dataset C.

In Figure 5, Figure 6 and Figure 7, (a) illustrates the overlapping area of the left image; (b) illustrates the overlapping area of the right image; (c) illustrates the road probability map of the overlapping area of the left image; (d) illustrates the road probability map of the overlapping area of the right image; (e) illustrates the *PRAs* of the overlapping area; and (f) illustrates the NCC cost of the overlapping area. Figure 5c, Figure 6c, and Figure 7c illustrate the road probability map of the overlapping area of the left image of Datasets A, B, and C, respectively. The larger the pixel value, the more likely it is to be determined as a road. Figure 5d, Figure 6d, and Figure 7d illustrate the road probability map of the overlapping area of the right image of Datasets A, B, and C, respectively. The larger the pixel value, the more likely it is to be determined as a road. Most of the roads in the orthoimages are extracted. In order to enhance the appearance of the road probability map, the stretch method of histogram equalize is used to adjust the value of the road probability map. Because of the differences between the overlapping area of the left and right image, the road probability maps obtained by them are also not the same, which are illustrated in (c) and (d) of Figure 5, Figure 6 and Figure 7. With the help of the Otsu method [41,42], the preferred road areas of the left and right image are determined by the road probability threshold for binarization, respectively. The final *PRAs* of the overlapping are defined from the preferred road areas of left and right image by Equation (4). In Figure 5e, Figure 6e, and Figure 7e, the white areas are the *PRAs* of the overlapping areas, which are the priority areas that the seamlines pass. Most roads of the overlapping area are included in the *PRAs*, which meets our requirement. Figure 5f, Figure 6f, and Figure 7f illustrate the NCC cost of the overlapping area of Datasets A, B, and C, respectively. The greater the brightness value of a pixel, the greater the NCC cost. The NCC cost is an effective method for assessing the difference of the pixel-level.

Figure 8 illustrates the experiments of seamline determination of the three different methods for Dataset A. Figure 9 illustrates the experiments of seamline determination of the three different methods for Dataset B. Figure 10 illustrates the experiments of seamline determination of the three different methods for Dataset C. The three methods were tested without a down-sampling strategy.

In Figure 8, Figure 9 and Figure 10, (a), (**c**), and (e) illustrate the seamlines determined using Dijkstra’s algorithm, OrthoVista and the proposed algorithm, respectively; (b), (d), and (f) illustrate the details of the white boxes in (a), (c), and (e), respectively. The ideal seamline should avoid crossing obvious objects, such as buildings, as much as possible and go along the objects which have small relief displacement, such as a road, river, grass, or bare land [6]. From the seamline detection results of the three data sets and especially the details of the white boxes, compared with the two previous methods, the seamlines determined by the proposed method mainly go along the roads where no significant differences exist.

In many related studies, it is difficult to find a general method for automated quantitative assessment of the seamline quality. Therefore, similar to the evaluation method in other relevant studies [2,6,7,25], the quantitative index applied in the proposed method is the number of times that seamlines cross obvious objects. The seamlines which have a smaller number of times are considered ideal seamlines. In order to compare the efficiency of the different algorithms fairly, all the algorithms were implemented without a down-sampling strategy and parallel computing strategy. The comparison results of the three different methods in the three groups of test data are shown in Table 2.

The coverage area of Dataset A is located in the central urban area of large cities, for which the spatial resolution is 0.5 m. There are many high-rise buildings, overpasses, residential areas, and other buildings in the image. Such objects are the objects which the ideal seamlines should bypass. The seamline determination by Dijkstra’s algorithm crossed five obvious objects, which is shown in Figure 8a. It crossed four bridges and an overpass, which is illustrated in Figure 8b. The seamline determination by the OrthoVista algorithm crossed nine obvious objects, which is shown in Figure 8c. Figure 8d shows that the seamline crossed several bridges and a buildings. Seamline determination by the proposed algorithm crossed one building, which is shown in Figure 8e. The details of the one crossed building are illustrated in Figure 8f. As shown in Figure 5e, during the optimization of the seamline at the semantic level, the proposed algorithms almost included the roads in *PRAs*. By giving the *PRAs* area a smaller weight, which is shown in Equation (7), the seamline mainly passes through the *PRAs*, and the final seamline is mainly along the roads, bypassing most of the areas with large relief displacement.

The coverage area of Dataset B is located in the suburbs area of large cities, for which the spatial resolution is 0.2 m. There is much agricultural land, woodland, and bare land in the image. There are some original villages and factories in it. A main road runs from north to south. The seamline determination by Dijkstra’s algorithm crossed six obvious objects, which is shown in Figure 9a. The details of the crossed buildings are illustrated in Figure 9b. The seamline determination by the OrthoVista algorithm crossed six obvious objects, which is shown in Figure 9c. The details of the crossed buildings are illustrated in Figure 9d. Seamline determination by the proposed algorithm crossed no obvious objects and was along the north–south main road, which is shown in Figure 9e. The details of the crossed buildings are illustrated in Figure 9f. As shown in Figure 6e, during the optimization of the seamline at the semantic level, the proposed algorithms almost included the roads in *PRAs*. By giving the *PRAs* area a smaller weight, which is shown in Equation (7), the seamline mainly passes through the *PRAs*, and the final seamline is mainly along the roads, bypassing most of the areas with large relief displacement.

The coverage area of Dataset C is located in the central urban area of a medium-sized city, for which the spatial resolution is 0.5 m. In the image, in addition to buildings and residential areas, there is much agricultural land and woodland. The seamline determination by Dijkstra’s algorithm crossed two obvious objects, which is shown in Figure 10a. It crossed two buildings, which is illustrated in Figure 10b. The seamline determination by the OrthoVista algorithm crossed 11 obvious objects, which is shown in Figure 10c. Figure 10d shows that the seamline crossed several buildings. Seamline determination by the proposed algorithm crossed two obvious objects, which is shown in Figure 10e. The details of the two crossed buildings are illustrated in Figure 10f. As shown in Figure 7e, during optimization of the seamline at the semantic level, the proposed algorithms almost included the roads in *PRAs*. By giving the *PRAs* area a smaller weight, which is shown in Equation (7), the seamline mainly passes through the *PRAs*, and the final seamline is mainly along the roads, bypassing most of the areas with large relief displacement.

In summary, due to the use of D-LinkNet, the roads were almost extracted and the *PRAs* were determined, which are shown in Figure 5e, Figure 6e, and Figure 7e. The seamline obtained by our method had the best result and passed through the less obvious objects and mainly went along the roads.

In practice, efficiency has to be taken into consideration. The processing time are recorded in the fourth column of Table 2. This shows that the seamlines determined by Dijkstra’s algorithm took around 644.025 s on average. OrthoVista’s method was better. It took around 14.333 s on average. The seamlines determined by the proposed method took around 22.784 +Δs on average. This processing time for the proposed method consists of two parts. The first part is the time required for extracting the road probability map using D-LinkNet, and the second part is the time required for seamline optimization. Δ represents the time required for training the network and extracting the road probability map. It took around 50 h. Regardless Δ, the proposed and OrthoVista methods are at the same level. Compared with Dijkstra’s and OrthoVista’s methods, Δ of the proposed method includes the network training time and road probability map extraction time. The network training time accounted for most of Δ. Although network training takes some time, once processed completely, it can be used to extract the road probability map for other data.

Figure 11 shows the experiments of seamline determination for Dataset A with different values of *w*. A quantitative comparison of seamlines determined by the proposed method for Dataset A with different values of *w* was conducted, as shown in Table 3. *w* is the weight for pixels in *PRAs,* which is shown in Equation (7). With such weight processing, this makes sure that the difference in the road area can be relatively small, so the seamlines will pass through roads as much as possible. In our paper, we suggested setting *w* to 0.001. According to Table 3, an acceptable seamlines determination result can be obtained by setting *w* to 0.001 for Dataset A.

## 4. Conclusions

In this paper, an automatic seamline determination method was presented for urban image mosaicking based on road probability map from the D-LinkNet neural network. The road probability map was used to improve the seamline determination. This method optimizes the seamlines at both the semantic and pixel level. At the semantic level, the *PRAs* are determined by binarizing the road probability map. In this step, most of the roads are included in the *PRAs*. At the pixel level, Dijkstra’s algorithm is adopted to find the final seamlines. To improve the efficiency, the minimum heap is adopted to store the graph in the form of adjacency lists and extract the minimum efficiently [7]. Three group data sets of aerial orthoimages with different ground resolutions located in different cities were used to test and validate the proposed method in this paper. The comparative experimental results show the advantages of the proposed method. Compared with two previous methods, the seamline obtained by the proposed method had the best result in that it passed through the less obvious objects and mainly followed the roads. In terms of the computational efficiency, the proposed method also has a high efficiency. Moreover, the proposed method can easily be applied to the seamlines network determination framework easily [3,46,47].

Nevertheless, the proposed method may be improved in the future as follows: (1) Road probability map generation by D-LinkNet may have a significant influence on the final seamline determination. Therefore, more training samples should be made to better train the D-LinkNet neural network. (2) The proposed algorithm can be applied to the seamline network determination framework [3,46,47] to construct a single seamless composite image automatically.

## Figures and Tables

**Figure 1 sensors-20-01832-f001:**
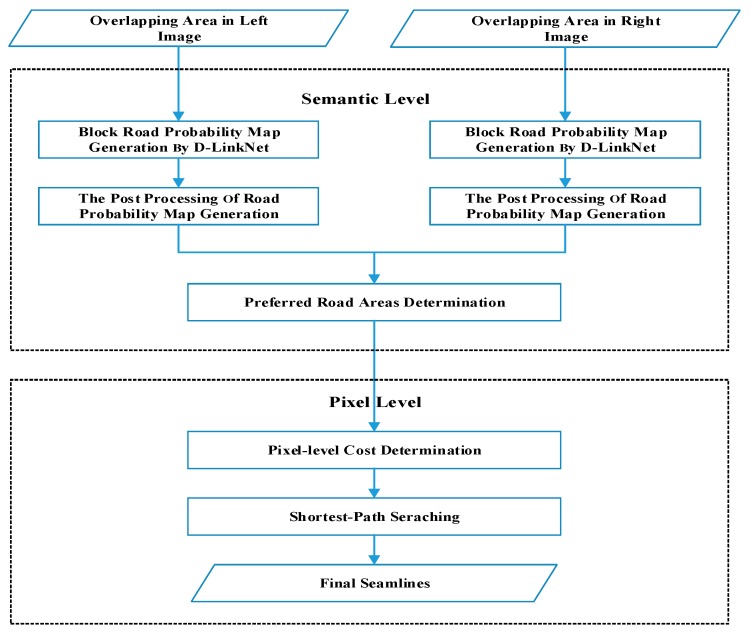
The process flow for the proposed method.

**Figure 2 sensors-20-01832-f002:**
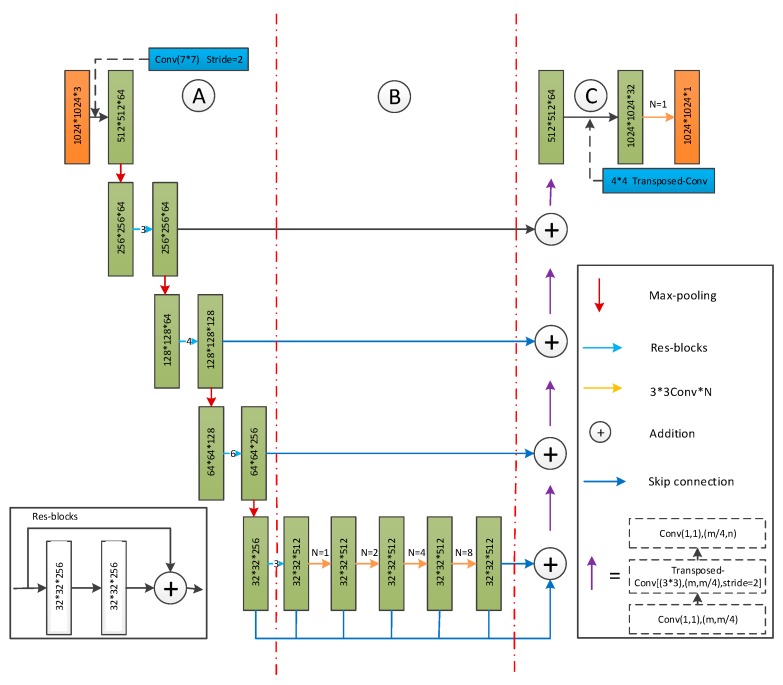
D-LinkNet architecture [34].

**Figure 3 sensors-20-01832-f003:**
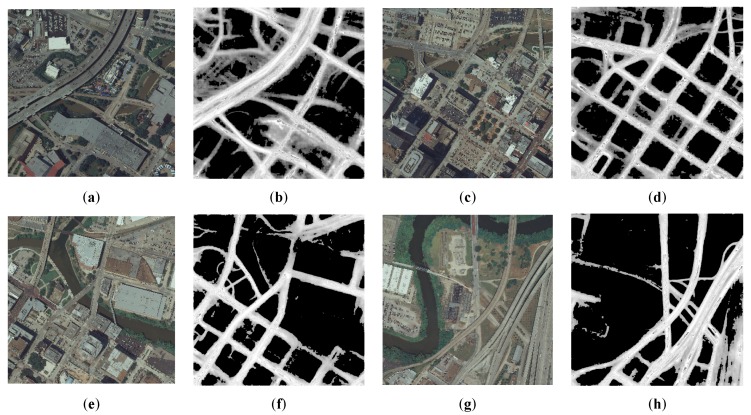
Road probability map generation by D-LinkNet: (**a**), (**c**), (**e**), and (**g**) are original images, (**b**), (**d**), (**f**), and (**h**) are road probability maps generation by D-LinkNet of (**a**), (**c**), (**e**), and (**g**).

**Figure 4 sensors-20-01832-f004:**
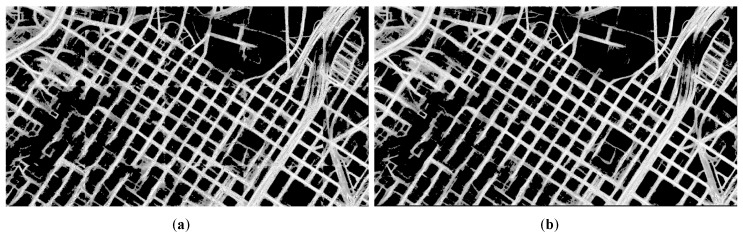
The stitched result of the road probability map obtained by different methods: (**a**) the stitched result of directly stitching method, (**b**) the stitched result of our stitching method.

**Figure 5 sensors-20-01832-f005:**
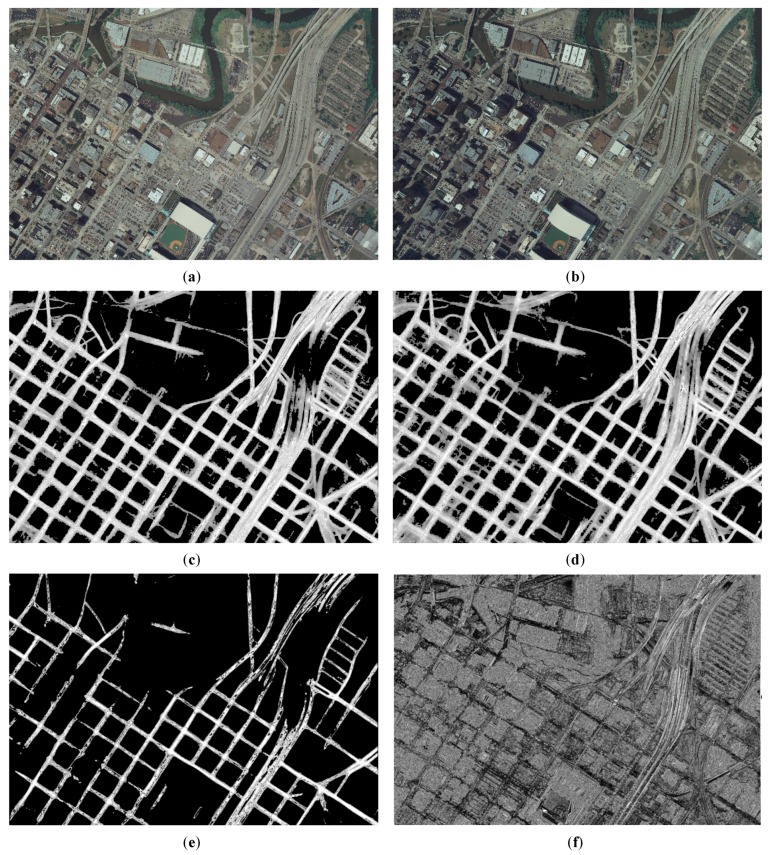
Experiments of preferred road areas (*PRAs*) determination for Dataset A: (**a**) the ovrlapping area of the left image, (**b**) the overlapping area of the right image, (**c**) the road probability map of the overlapping area of the left image, (**d**) the road probability map of the overlapping area of the right image, (**e**) the *PRAs* of the overlapping area, and (**f**) the Normalized Cross Correlation (*NCC*) cost of the overlapping area.

**Figure 6 sensors-20-01832-f006:**
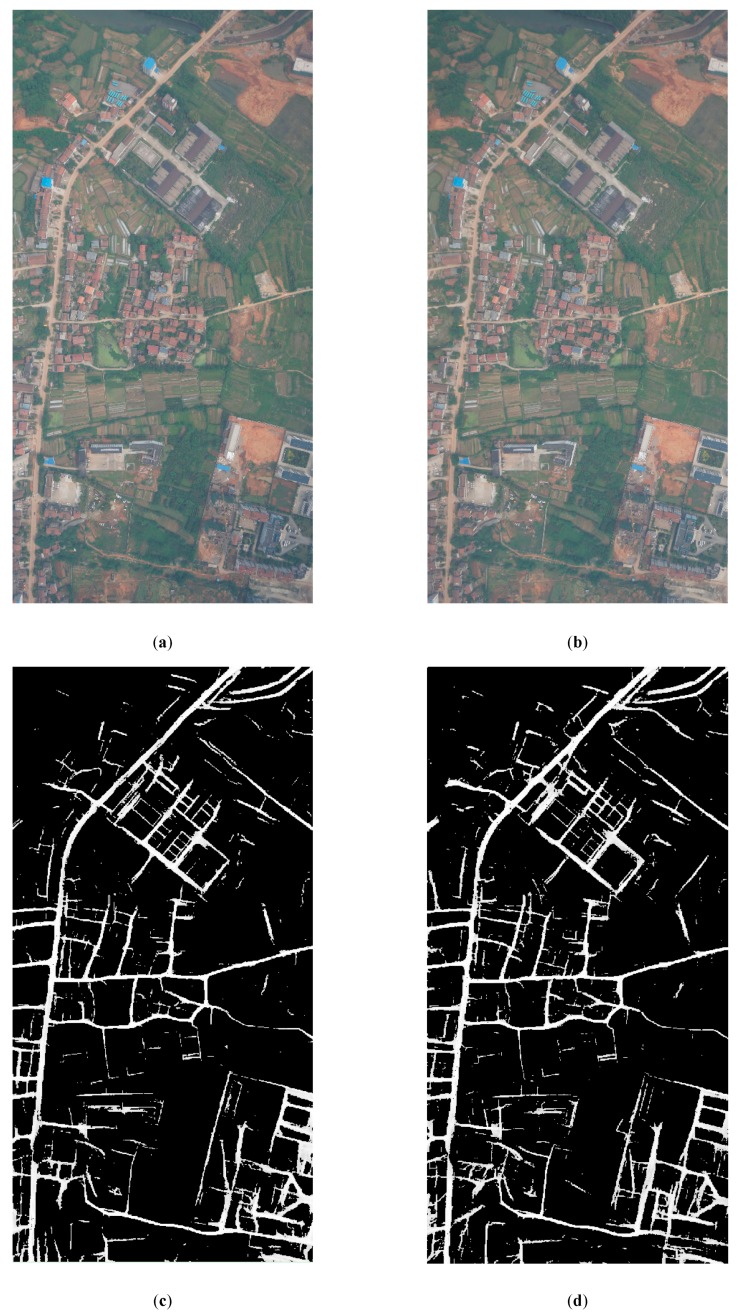
Experiments of preferred road areas (*PRAs*) determination for Dataset B: (**a**) the overlapping area of the left image, (**b**) the overlapping area of the right image, (**c**) the road probability map of the overlapping area of the left image, (**d**) the road probability map of the overlapping area of the right image, (**e**) the *PRAs* of the overlapping area, and (**f**) the Normalized Cross Correlation (*NCC*)cost of the overlapping area.

**Figure 7 sensors-20-01832-f007:**
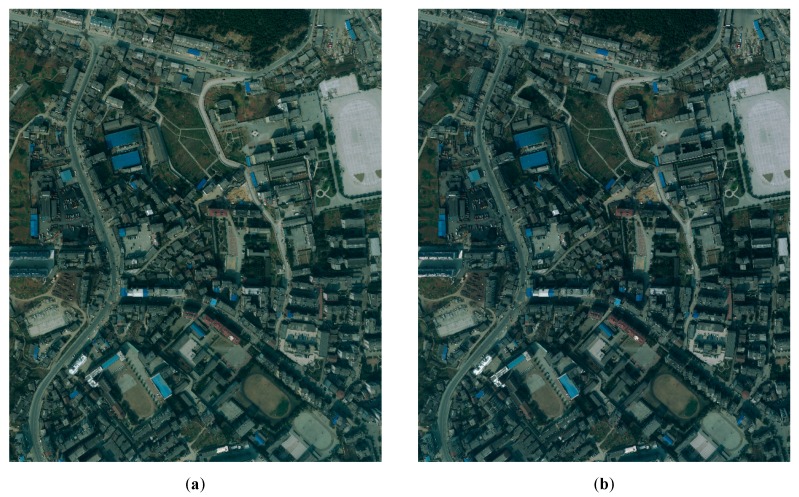
Experiments of preferred road areas (*PRAs*) determination for Dataset C: (**a**) the overlapping area of the left image, (b) the overlapping area of the right image, (**c**) the road probability map of the overlapping area of the left image, (**d**) the road probability map of the overlapping area of the right image, (**e**) the *PRAs* of the overlapping area, and (**f**) the Normalized Cross Correlation (*NCC*)cost of the overlapping area.

**Figure 8 sensors-20-01832-f008:**
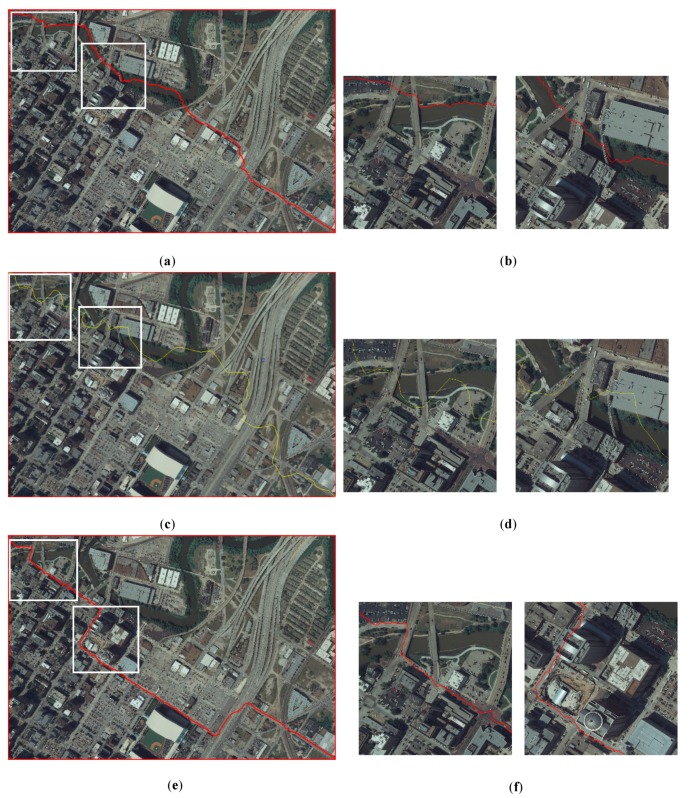
Experiments of seamline determination for Dataset A: (**a**) seamline determined by Dijkstra’s algorithm, (**b**) details of the white box in (**a**), (**c**) seamline determined by OrthoVista, (**d**) details of the white box in (**c**), (**e**) seamline determined by the proposed method, and (**f**) details of the white box in (**e**).

**Figure 9 sensors-20-01832-f009:**
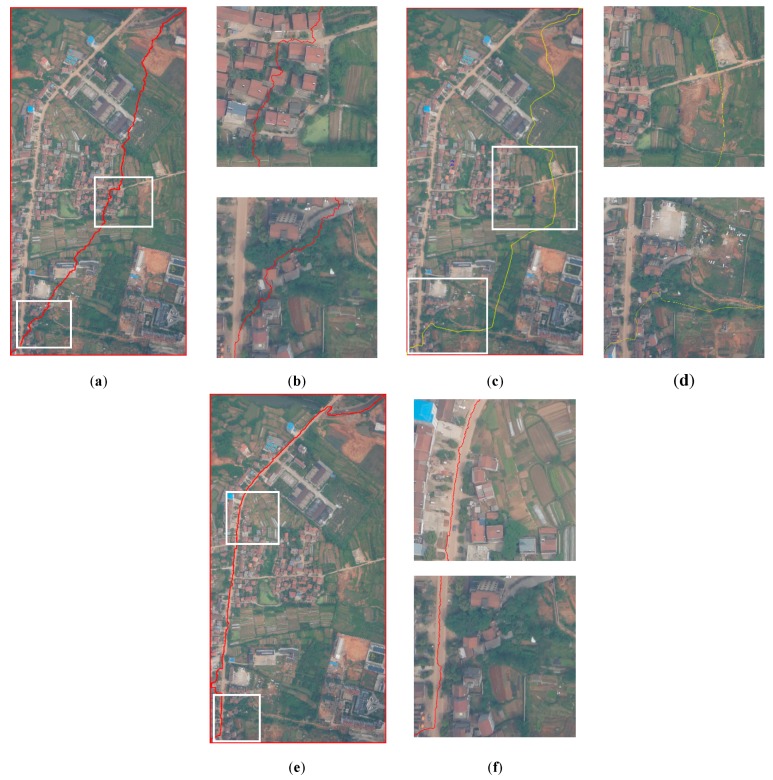
Experiments of seamline determination for Dataset B: (**a**) seamline determined by Dijkstra’s algorithm, (**b**) details of the white box in (**a**), (**c**) seamline determined by OrthoVista, (**d**) details of the white box in (**c**), (**e**) seamline determined by the proposed method, and (**f**) details of the white box in (**e**).

**Figure 10 sensors-20-01832-f010:**
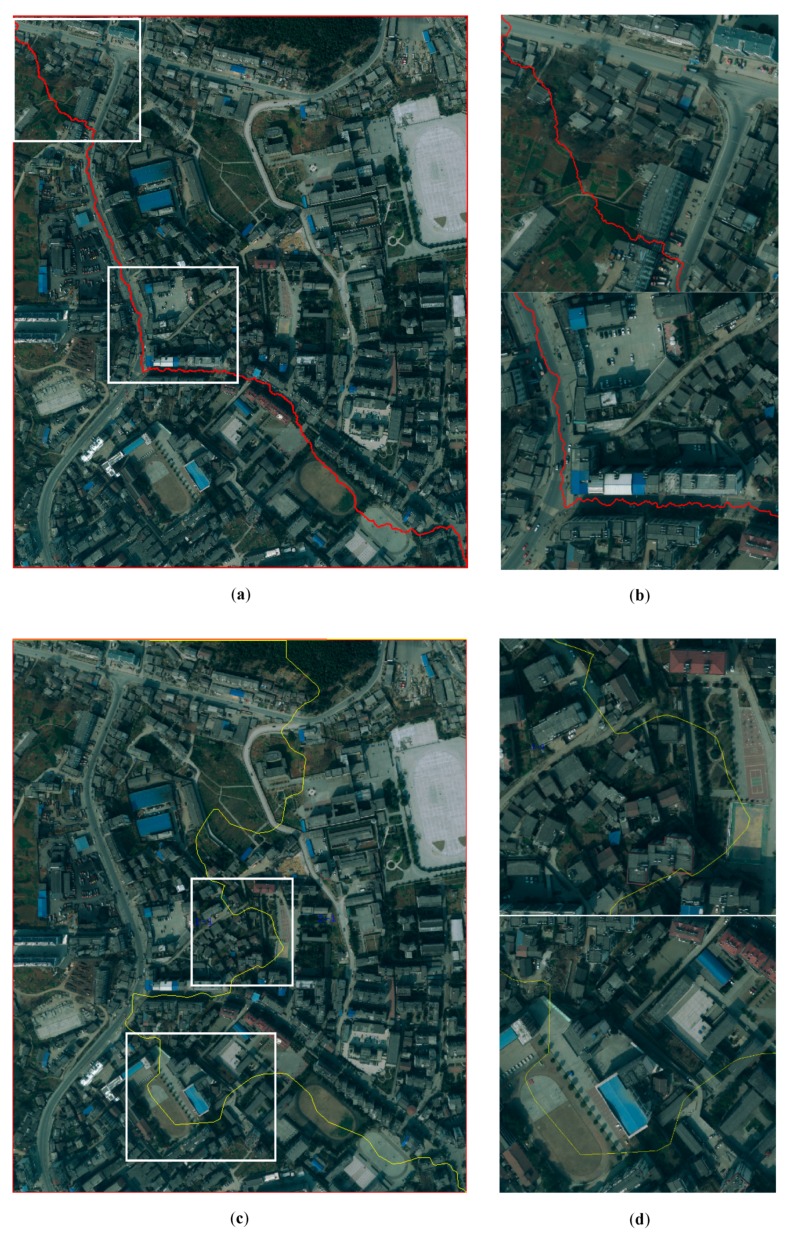
Experiments of seamline determination for Dataset C: (**a**) seamline determined by Dijkstra’s algorithm, (**b**) details of the white box in (**a**), (**c**) seamline determined by OrthoVista, (**d**) details of the white box in (**c**), (**e**) seamline determined by the proposed method, and (**f**) details of the white box in (**e**).

**Figure 11 sensors-20-01832-f011:**
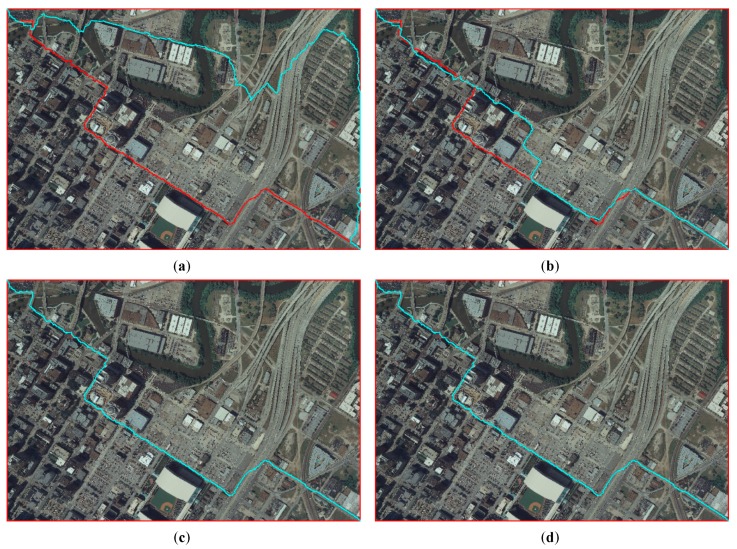
Experiments of seamline determination for Dataset A with different values of *w*: (**a**) *w* is 0.001 (red line) and *w* is 0.1 (cyan line), (**b**) *w* is 0.001 (red line) and *w* is 0.01 (cyan line), (**c**) w is 0.001 (red line) and *w* is 0.0001 (cyan line), and (**d**) *w* is 0.001 (red line) and *w* is 0.00001 (cyan line).

**Table 1 sensors-20-01832-t001:** Basic information of the datasets.

Dataset	Image Resolution	Imaging Size	Coverage Features
Dataset A	0.5	3030 × 2067 × 3	Central area of a big city
Dataset B	0.2	2438 × 4824 × 3	Suburb of a big city
Dataset C	0.3	2212 × 2693 × 3	Central area of a medium-sized city

**Table 2 sensors-20-01832-t002:** Comparison of previous methods with the proposed method.

Dataset	Method	Number of Obvious Objects Passed Through	Processing Time (s)
1	Dijkstra’s	5	329.770
	OrthoVista	9	13.000
	Proposed	1	21.387+Δ
	Dijkstra’s	6	1033.976
2	OrthoVista	6	21.000
	Proposed	0	38.353+Δ
	Dijkstra’s	2	568.329
3	OrthoVista	11	9.000
	Proposed	2	8.612+Δ

**Table 3 sensors-20-01832-t003:** Comparison of the proposed method with different values of *w* for Dataset A.

The Values of *w*	Number of Obvious Objects Passed Through	Processing Time (s)
0.1	11	24.522+Δ
0.01	4	18.602+Δ
0.001	1	21.387+Δ
0.0001	1	25.461+Δ
0.00001	1	25.491+Δ

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
