# Peer review of "Automatic Seamline Determination for Urban Image Mosaicking Based on Road Probability Map from the D-LinkNet Neural Network"

_sensors, 2020, doi:10.3390/s20071832_

Round 1

Reviewer 1 Report

The paper presents a pipeline for seamline determination by combining different known algorithms, including D-LinkNet for road probability map generation, post-processing, Otsu thresholding method, Normalized Cross Correlation,…

The paper technically sounds. However, there are several issues:

   - The description of the method is not clear, not detailed. In Section 2.2, if the reader understood, the authors used the known Otsu method to estimate the  T1 and T2 thresholds. However, in the text, “…we use the Otsu to estimate T1. The proposed method estimates…”. What is the “proposed method” for threshold estimation? How was T2 estimated? Some illustrations of this step (using Otsu) should be included.

   - In Section 2.3.1, why were “5x5 windows” used? Did the authors try other window’s size? What is the performance?

  - In Section 2.3.2, the authors used the similar idea as in in [7], more discussion/comments should be included (for example, the complexity). Minor: the formula should be re-edited.

  - In Experiments, some parameters were not clearly presented. For example, “the proposed method was performed with w=0.001. In practice, the value w is determined by our experience”. So, what is the value of w? what is the experience carried out for determining w?

  - It is difficult to follow the Experiment results. For example, line 326 “Figure 5, 7, 9…”, the reader has to change 4-5 pages to look at the Figure. The presentation should be improved to make it easier to read the paper. Also there are a lot of typos here: 

      lines 326, 327, ”illustrates” 

      line 342, “white area are…”

      lines 348-350: difficult to understand.

      line 382: “the evaluation method adopted of the proposed method…”: not clear

      Table 2: “what is the value of delta”

There are many typos in the paper (in Figure 1, “Aera”) and English quality is not very good, this makes the paper difficult to follow. The authors should revise the paper more carefully. More justification of the algorithm and discussion of experiment results should be included.

Reviewer 2 Report

The authors present a methodology and a workflow for the automatic determination of a seamline at urban areas based on Road probability map acquired the D-LinkNet neural network.

They authors performed a thorough investigation of the existing approaches and methodologies. They present very clearly their approach and methodology, while their experiments are well designed covering areas with different characteristics.

Their results seem promising and support their conclusions.

The manuscript needs re writing with respect to the English language.  Some examples below

Line 24 instead of “final seamlines is” should be “final seamlines are”

Line 72 “used to calculation the photometric” “used to calculate”, it is not clear what photometric refers too.

Line 105 “dijksta” should be “Dijkstra”

Line 188 Figure 1 should be Figure 2

Regarding the NN training it is not clear in the manuscript if they perform the training once and then apply the NN to the different areas or they must train the network for each area. It is my understanding that the training occurs once, but they should clarify that.

Regarding the evaluation of the different methods they might consider alternative measures apart the number of obvious objects, for example the count of pixels that are passing through obvious objects, or compare the results of the mosaicked images around the seamlines and report the number of artifacts created by a “wrong” seamline.

Round 2

Reviewer 1 Report

In the revised version, almost issues were addressed.

Minor details could be improved: for example, “2 obvious objects” -> “two obvious objects” (in general with a number smaller than 10, we write it in letters. Of course, we write “Figure 8”, “Table 2”, not “Table two” since “2” is the “name” of the table).

The English quality was improved but the formula editing can still be improved.

Line 195, “Figure2”

Line 215, “Fig. 4(a)” (use Figure or Fig., not both)

Line 325, “w” was written in two different ways (italic and normal)

To conclude, almost technique issues were addressed and English quality was improved. However, text editing should be improved.